# Cryptanalysis of Two Conditional Privacy Preserving Authentication Schemes for Vehicular Ad Hoc Networks

Ahmad Mohamad Kabil [1], Heba Aslan [1] and Marianne Azer [2],*

1   School of Information Technology & Computer Science, Nile University, Cairo 3247010, Egypt;
    a.mohamad2116@nu.edu.eg (A.M.K.); haslan@nu.edu.eg (H.A.)
2   National Telecommunications Institute, Cairo 3650108, Egypt
*   Correspondence: mazer@nu.edu.eg

**Abstract:** Conditional Privacy Preserving Authentication (CPPA) schemes are an effective way of securing communications in vehicular ad hoc networks (VANETs), as well as ensuring user privacy and accountability. Cryptanalysis plays a crucial role in pointing out the vulnerabilities in existing schemes to enable the development of more resilient ones. In 2019, Zhang proposed a CPPA scheme for VANET security (PA-CRT), based on identity batch verification (IBV) and Chinese Remainder Theorem (CRT). In this paper, we cryptanalyze Zhang's scheme and point out its vulnerability to impersonation and repudiation attacks. In 2023, Zhang's scheme was cryptanalyzed by Tao; however, we point out flaws in Tao's cryptanalysis due to invalid assumptions; hence, we propose countermeasures to Tao's attacks. Furthermore, in 2021, Xiong proposed a Certificateless Aggregate Signature (CLAS) scheme which is also cryptanalyzed in this paper. Finally, we analyze the causes and countermeasures by pointing out the vulnerabilities in each scheme that enabled us to launch successful attacks and proposing changes that would fortify these schemes against similar attacks in the future.

**Keywords:** vehicular ad hoc networks; conditional privacy preserving authentication schemes; identity based batch verification; certificateless aggregate signature schemes



## 1. Introduction

Cooperative Intelligent Transportation Systems (C-ITS) are systems that employ inter-vehicular communication to provide a safer and more comfortable driving experience to road users, by sensing the surrounding road environment and pooling their collective data to make reliable decisions [1]. A C-ITS is composed of vehicles, roadside infrastructure and a traffic management authority. Therein, communication takes place in an ad hoc manner through self-organized networks called vehicular ad hoc networks (VANETs). The development of C-ITS has garnered substantial attention from the international community. The American government identified intelligence and connectivity as two of its core strategies, Japan recently released a report to share guidelines of autonomous driving, and Korea released a long term development plan aiming to achieve intelligent transportation systems nationwide by 2040 [2]. While cellular communication can be used in specific cases, the main standards of vehicular communication are the dedicated short-range communications (DSRCs) standard developed in the US, and the Intelligent Transportation System (ITS-G5) protocol developed by the European Telecommunications standards institute (ETSI) [1]. While both standards are fundamentally based on the IEEE 802.11p access layer developed for vehicular networks, a competing alternative called C-V2X has recently emerged, satisfying the low latency requirements as well as supporting high vehicle densities and/or speeds [3]. More efficient and scalable alternatives have recently come to light, like 802.11bd and New Radio V2X [4], striving to satisfy stringent requirements of low latency, reliability, a maximum packet error rate of 10% with a minimum transmission radius of 300 m. To ensure proper functioning of C-ITS applications, it is required that large amounts of data

are continuously exchanged. Owing to the life-critical nature of this data; authentication, confidentiality, integrity, privacy and revocability are non-negotiable security requirements to ensure proper functioning of VANETs.

To that effect, a multitude of works have strived to create Conditional Privacy Preserving Authentication (CPPA) schemes, which can be broadly classified into certificate-based schemes/Public Key Infrastructure (PKI) schemes, Identity-based schemes (IB), Group-signature-based schemes (GS) and Certificateless schemes (CL). In order to support scenarios of large vehicular densities, techniques to verify a batch of digital signatures simultaneously have been developed, leading to extensions of IB and CL schemes into identity-based batch verification schemes (IBV) and certificateless aggregate signature schemes (CLAS), respectively. While researchers are quick to claim that their schemes are provably secure, the literature proves otherwise since many contributions involve pointing out weaknesses in previous schemes, which is a necessary pre-requisite to their development. This highlights the importance of works that cryptanalyze prior works and point out points of vulnerability, which shall undoubtedly enhance the resilience of future schemes.

In 2019, Zhang et al. proposed an IBV scheme, PA-CRT [5], based on the Chinese remainder theorem, and in 2023, PA-CRT was cryptanalyzed by Tao et al. [6]. However, we propose a simple modification to circumvent Tao's attacks and conduct our own cryptanalysis of the PA-CRT scheme in this paper. In 2021, Xiong et al. proposed a CLAS scheme, CPPA-D [7], which was cryptanalyzed by Shim [8] in 2023. The authors introduced an improved scheme [9] in 2023, which we also cryptanalyze in this paper. Accordingly, in this paper, our main contributions are as follows:

- We show that Zhang's IBV scheme [5] is vulnerable to impersonation attacks where any member of a VANET can easily obtain the private information of other members and generate and sign messages on their behalf. We discuss causes and propose general improvements to the scheme to mitigate this attack. We also show that the same scheme [5] is vulnerable to repudiation attacks where malicious users can send messages with false information using fake identities and escape retribution accordingly. This encourages them to send false information to suit their own purposes and easily escape accountability in case their messages are reported to a trusted authority. Finally, we point out the deficiencies in Tao's [6] cryptanalysis of Zhang's IBV scheme [5].

- We show that Xiong's CLAS scheme [9] is vulnerable to bogus information attacks since partial private keys and pseudo-identities are not adequately verified and can be replaced by the sender of the message and still perform successful verification at the receiver's end. We also show that the scheme is vulnerable to the same repudiation attack that afflicts Zhang's scheme and point out a mistake in the batch verification equation.

The rest of this paper is set out as follows. Section 2 gives a general introduction to VANET security requirements, Section 3 presents related works, in which we succinctly describe earlier contributions of IBV and CLAS schemes and their development in recent years, Section 4 presents mathematical preliminaries, Section 5 presents our crypt analysis of Zhang's scheme, Section 6 presents our crypt analysis of Xiong's scheme and Section 7 presents our concluding remarks. A list of notations can be found in Table 1.

**Table 1.** Notations and terms.

| Notation | Term |
| --- | --- |
| CA | Central Authority |
| CRT | Chinese Remainder Theorem |
| C-ITS | Cooperative Intelligent Transportation Systems |
| CL | Certificateless (Schemes) |
| CLAS | Certificateless Aggregate Signature (Schemes) |
| CPPA | Conditional Privacy Preserving Authentication |

**Table 1.** *Cont.*

| Notation | Term |
| --- | --- |
| DSRC | Dedicated Short Range Communications |
| EC(C) | Elliptic Curve (Cryptography) |
| ECDLP | Elliptic Curve Discrete Logarithm Problem |
| GS | Group Signature |
| IBC | Identity Based Cryptography |
| IBV | Identity Based Batch-Verification (Schemes) |
| KGC | Key Generation Center |
| MAC | Message Authentication Code |
| OBU | On Board Unit |
| PKI | Public Key Infrastructure |
| PID | Pseudo Identity |
| PWD | Password |
| RID | Real Identity |
| RSU | Roadside Unit |
| TA/TTP | Trusted Authority/Trusted Third Party |
| TPD | Tamper Proof Device |
| TRA/TRM | Tracing Authority/Tracing Manager |
| VANET | Vehicular Ad hoc Network |
| V2I | Vehicle-to-Infrastructure |
| V2V | Vehicle-to-Vehicle |
| V2X | Vehicle-to-Everything |

## 2. VANET Security Requirements

In VANETs, vehicles communicate regularly with other vehicles as well as road infrastructure (RSUs) in order to exchange periodic safety messages. This communication is necessarily subject to stringent security requirements to preserve the safety and well-being of VANET users. In order to receive a message, it must be ensured that the vehicle has the capacity to receive and process the message (availability), furthermore, the vehicle must be able to verify that the message was sent by a legitimate participant (authentication) and contains valid information that is temporally relevant (integrity). From the sender's perspective, it must be ensured that they are not vulnerable to any revelation of their private information (privacy) except by relevant authorities so that they may be held accountable in case of a dispute (traceability) and cannot deny having sent the message (non-repudiation). The features of privacy and traceability combined together constitute what has come to be one of the fundamental objectives of VANET security systems: conditional privacy. Conditional privacy stipulates that messages are transmitted anonymously by all participating members in VANETs and that all external parties, except law enforcement authorities, are incapable of discerning the identity of the sender of any message. Finally, one of the most important security objectives in any communication network is to guarantee that none but the intended recipient has the ability to infer sensible information from the contents of a message (confidentiality). While the importance of confidentiality in VANETs has been downplayed under the premise that everyone deserves access to the contents of basic safety messages that vehicles periodically transmit [10], others have pointed out that confidentially is necessary in paid services and have constructed symmetric key exchange schemes accordingly [11].

In order to guarantee the abovementioned security objectives, a combination of cryptographic approaches and intrusion detection approaches is used. Cryptographic protection can be construed as the first line of defense that offers preemptive protection against possible attacks, but once an attack occurs, intrusion detection may be employed to take necessary measures of damage control. This paper is primarily concerned with cryptographic techniques employed to preserve secure communications in VANETs. Intrusion detection in VANETs is also being studied extensively in the contemporary literature and reviews on contributions can be found in [12–14].

### 2.1. Attacks on Availability

The objective of availability in VANETs is threatened by denial-of-service attacks and spamming attacks, in which a malicious user sends an excessive number of spam messages to a vehicle or RSU in order to prevent them from being able to access valid messages containing critical information. The attacker in this case is considered malicious since they do not derive any personal benefits from this attack. They are also considered active since they are not restricted to obtaining information but are instead instigating material change within the network. Finally, they could be insiders or outsiders since both have the ability to jam the network with spam messages. This type of attack is detected by machine-learning-based intrusion detection and could be mitigated by frequency hopping and channel switching techniques. Other threats to the objective of availability include blackhole and greyhole attacks in which malicious VANET users receive safety messages from other vehicles but refrain from transmitting all packets (blackhole) or selected packets (greyhole), and hence, legitimate users do not receive the required information. Attackers in this case are considered malicious since they do not reap personal rewards, active since they affect the network and must be insiders in order to receive the messages to begin with. Finally, the threat of a malware attack very imminently affects the objective of availability since malware afflicts VANET system components like OBUs and RSUs, rendering them unable to send or receive messages.

### 2.2. Attacks on Authentication

The objective of authentication in VANETs is threatened by a variety of impersonation attacks, in which an attacker claims a false identity in order to illegally obtain confidential information and perhaps send fake messages as well. Different impersonation attacks include a masquerading attack, in which the attacker assumes the identity of a legitimate node and sends and receives messages under this guise; a man-in-the-middle attack, in which the attacker impersonates two legitimate users having a private conversation and addresses each on behalf of the other, thus obtaining sensitive private information about both [15]; and a certificate replication attack, in which the attacker duplicates the identities of legitimate users in order to avert the trusted authority (TA) that supervises the VANET communications. Other more elaborate forms of impersonation attacks include a sybil attack, where, for example, the attacker generates 100 different identities and sends the same fake message of an upcoming road congestion through all of them, thus reinforcing the validity of this synthesized illusion and encouraging all legal participants to switch course to avoid the congestion, hence freeing up the road for this attacker. Other threats to authentication include attacks in which location information of the sender is falsified, like tunneling/wormhole attacks, in which the attacker exploits an alternative communication channel to create the illusion of being a neighboring node, whereas in fact, they are operating remotely; a similar example is a GPS spoofing attack in which the attacker falsifies their GPS information. Attackers in these cases are usually insiders to the VANET network, they could operate rationally or maliciously depending on whether they intend to simply cause harm or reap personal benefits, as in the example of the Sybil attack, or even more sinisterly launch these attacks for the purpose of terrorism. They could also be active or passive depending on whether they simply seek to extract useful information about other users for future exploitation or whether they seek to immediately send false messages within the network. These attacks are mitigated cryptographically using digital signatures, which are bound to the identity of the user using digital certificates [15], as well as message authentication codes (MACs) which are keyed hash functions used to preserve message integrity and authentication.

### 2.3. Attacks on Integrity

The objective of message integrity is strongly intertwined with that of authentication, since in very simplistic terms, if an attacker were to send false information, they would probably assume a false identity to do so in order to avoid accountability. The same

cryptographic primitives used to guarantee authentication (digital signatures and message authentication codes) also implicitly guarantee the integrity of the message since any change in the contents of the message would immediately change the associated digital signature/ authentication code. In [16], an alternative way to mitigate attacks on message integrity was suggested, namely, through verification by correlation, where data from multiple sources are correlated using reputation-based systems in order to ensure the validity of received messages. Attacks on message integrity include message tampering, broadcast tampering and bogus information attacks, all of which involve injecting false data into the message and altering its contents. Other attacks on integrity include the replay attack where valid data are transmitted fraudulently under the premise of being temporally relevant, when it actually concerns a previous irrelevant time slot. Defense against replay attacks is achieved by appending a timestamp to each digitally signed message.

### 2.4. Attacks on Conditional Privacy

The objective of conditional privacy is threatened by attacks in which user data (specifically location data) are extracted from their sent messages, to be later used in exploiting them if valid data are inferred about locations which the user frequently visits. Protection against such attacks is achieved using privacy preserving techniques which include anonymous certificates (pseudo-identities), or group signatures, in which members within a VANET group anonymously sign messages on behalf of the group and other members can verify the signatures using the public key of the group. A group manager/opener is the only entity that has the authority to divulge the identity of the group member that signed the message in case of dispute, but other members can only verify that the signer is one of the legitimate group members without ascertaining which member of the group that is.

### 2.5. Attacks on Non-Repudiation

The objective of non-repudiation is inherently achieved in any system that employs public key cryptography where users sign messages using their private key and broadcast their public key to enable other users to authenticate. Since the private key of any user is inextricably linked to their public key, this implicitly guarantees that the signer of the message did in fact sign the message, so that they cannot deny having signed it later on. While symmetric key cryptography (in which both parties have equal access to the same encryption/decryption key) is advantageous over public key cryptography (in which users retain their private keys and only share public keys) in terms of efficiency and speed, it is only the latter that can guarantee non-repudiation which renders public key cryptography indispensable in any network security system, despite its relative complexity and high computational cost. The most common attack is a repudiation attack where the attacker can find means by which to escape accountability for their sent messages, this is achieved if the attacker exploits weaknesses in a scheme in order to broadcast messages using fake identities that do not undeniably condemn them.

### 2.6. Attacks on Confidentiality

Finally, the objective of confidentiality is achieved by simply encrypting messages before transmission. While the importance of confidentiality has been downplayed in many contributions, other contributions like [17] focus exclusively on cryptographically ensuring confidentiality in VANETs and devising lightweight ciphers to achieve that. Attacks on confidentiality include eavesdropping attacks, which are passive rational attacks in which the attacker listens in on private conversations to extract useful information for personal gain, and traffic analysis attacks in which attackers analyze the frequency of transmitted messages in order to derive purposeful information.

Having detailed the various security objectives of VANETs, it is also important to mention another important objective without which VANET systems fail to operate reliably, namely, that of efficiency. Vehicles periodically send safety messages every 0.3 s, so for a high-density area, a vehicle would receive a huge number of messages every 0.3 s that need

to be verified before the next batch comes in, hence batch verification (the simultaneous verification of a multiple of messages) is an essential component of VANET security.

In Table 2, we provide a summary of different VANET security requirements, threats and protection approaches.

**Table 2.** Summary of VANET security requirements, threats and protection approaches.

| Requirement | Description | Attacks | Protection |
|---|---|---|---|
| Availability | Timely arrival of critical information | Denial of service, spamming<br>Blackhole, greyhole<br>Malware | Frequency hopping<br>Channel switching<br>IDS |
| Authentication | Messages are sent by a legitimate user | Impersonation<br>Sybil<br>Wormhole, tunneling | Digital signatures<br>MAC tags<br>IDS |
| Integrity | Message contents have not been modified | Message/broadcast tampering<br>Replay | Digital signatures<br>MAC tags |
| Conditional Privacy | Only authorized entities have access to user identity | Location extraction attacks<br>ID disclosure attacks | Pseudonyms<br>Group signatures |
| Non-repudiation | Sender cannot deny having sent the message | Repudiation attacks | Public key cryptography |
| Confidentiality | Third party cannot extract meaningful information from any message | Eavesdropping<br>Traffic analysis | Encryption |

Having outlined the essential security requirements of VANETs, we discuss the historical development of IBV and CLAS schemes in the following section.

## 3. Related Work: IBV and CLAS Schemes

In this section, we detail the development of IBV schemes and CLAS schemes; from their theoretical underpinnings to their first concrete instantiations in the context of VANET and finally to the recent developments.

### 3.1. IBV Schemes

Identity-based batch verification (IBV) schemes are schemes that involve a Key Generation Center which issues public and private keys for all members and acts as the primary entity of authentication.

### 3.1.1. History of IBV

The main advantage of IB schemes is that they avoid the complexity of managing certificate revocation. In 1985, it was proposed in [18] that a TTP/CA would generate a public key for the network member (Alice) using an aspect of her identity. Then, using its private key, the TTP would generate a private key for Alice that would be a function of both the TTP's private key and Alice's public key $ID_A$. If Bob wanted to send a message to Alice, Bob would use both $ID_A$ and the TTP's public key to encrypt the message. In fact, under IBC, it is feasible that Bob sends a message to Alice before Alice receives her private key (decryption key) from the TTP and that Alice will only be allowed to read the received message if she is still a legitimate member of the network, hence circumventing the tedious task of certificate revocation. By including a timestamp along with $ID_A$, the TTP would only allow Alice to obtain a decryption key if $ID_A$ has not expired before Bob's message was sent. In [19], 16 years after the initial proposition in [18], the first practical IBC-based scheme was proposed, which used bilinear pairings in its implementation.

### 3.1.2. IBV in the Context of VANETs

The first application of IB based cryptography to VANETs was achieved by C. Zhang et al. [20] in 2008. They proposed IBV, an identity-based authentication scheme

that supports batch verification. In their scheme, pseudonymity is achieved by changing the PID with each message, which is generated using the master private key of the system that is securely stored inside the TPD of the vehicle. Pseudo-IDs conceal the real ID of the vehicle by El Gamal Encryption over ECC, and only the TTP can retrieve the real identity $RID$ from the pseudo identity $(ID_1, ID_2)$ as follows: $RID = ID_2 \oplus sID_1$, where $s$ is the master private key of the system. RSUs receive requests from vehicles and either handle them locally or forward them to the TTP depending on the nature of the request; they also monitor and summarize the traffic situation at their location and report it to the traffic control center. The signature on the message is a one-time identity-based signature with a length of 21 bytes, and the pseudo identity is 42 bytes, so the total overhead is much smaller than certificate-based schemes which involve the 125-byte-sized IEEE 1609.2 certificate. However, the scheme assumes ideal TPDs in vehicles, which is unreasonable to assume since these are expensive to install in every vehicle, and if side channel attacks are successful in recovering the system master key from the TPD, then the whole network is compromised. The verification cost of a single signature is one map-to-point hash, one multiplication and three bilinear pairings, which is relatively high, but the batch verification cost for $n$ signatures is $n$ multiplications, $n$ map-to-point hashes and only three pairings as well, which presents substantial efficiency gains compared to sequential verification of individual signatures. In this scheme, during batch verification, an invalid batch is automatically dropped, which is inefficient. Accordingly, they extend their scheme in 2011 [21], which employs a group testing technique for false signature detection prior to batch verification. Noteworthy is the fact that in their earlier paper in 2008, their scheme was based on the assumption that vehicles send messages to the RSU, which verifies them and subsequently broadcasts to other vehicles in the region, but they modified it in 2011 so that vehicles immediately received and verified transmitted messages by other vehicles without the intermediation of the RSU. C. Zhang's IBV scheme is of particular interest since it elicited multiple response papers in future years, which we shall detail subsequently in this section.

### 3.1.3. Criticisms of C. Zhang's Scheme and Improvements

In 2011, Chim et al. [22] proposed SPECS, a software-based solution that does not rely on TPDs and uses two shared secrets to satisfy the privacy requirements in VANETs. They proposed RSU-based message verification using two bloom filters to achieve a low overhead, where one stores valid signatures and the other stores the invalid ones. The scheme also employs binary search techniques during batch verification to identify the invalid signature in a batch. Their scheme also facilitates RSU-managed groups which vehicles can join after being authenticated. They assume that the TA is always online and fully trusted and propose redundant TAs with identical databases to avert a single point of failure threats. During the system setup, the TA generates a system master key and generates and publishes the corresponding system parameters. During registration, the vehicle receives its real ID (RID) and password (PWD), then receives new pseudo-IDs from the TA whenever it passes a new RSU. The pseudo-IDs are unlinkable since they involve hashing the real identity with a random nonce, which the vehicle encrypts with the TA's public key before transmitting through the RSU. SPECS achieves greater efficiency than IBV by using two bilinear pairings instead of three, and two bloom filters with opposite indications are employed simultaneously to reduce the false positive rates of traditional bloom filters by 89.6%. The authors also perform cryptanalysis of Zhang's IBV scheme and point out its overdependence on ideal TPDs in vehicles as well as its lack of consideration of the V2V scenario. The latter comment was actually modified in Zhang's extension of the scheme in 2011, as we pointed out earlier. The authors point out that IBV suffers from privacy violation, anti-traceability and impersonation attacks; we shall detail their point of view and explain why we disagree with them. In SPECs, it is incorrectly stated that in IBV, any malicious vehicle can obtain the real ID of an honest vehicle from its pseudo-id (using the same extraction method we outlined above), but this assumes that a malicious user has

easy access to the master secret key of the system that is securely stored in the TPD of the vehicle, which is not a practical assumption to make, since the IBV scheme assumes ideal TPDs that self-destruct if it recognizes attempts at infiltration. If the assumption in SPECs is true, this would intuitively lead to privacy violation followed by possible impersonation attacks as well. Finally, they criticize IBV because of its anti-traceability, assuming that a malicious vehicle can change its own real ID to any random value to prevent the TA from holding it accountable, which again assumes easy and convenient access to the values securely stored within the TPD as well as the ability to overwrite them. We conclude by stating that the only justified criticism of the IBV scheme is its over-reliance on ideal TPDs stored in each vehicle, which is both too expensive and too risky to implement in reality, but we do not concur with the criticisms stated in SPECs.

In 2012, Shim et al. [23] proposed CPAS which tackles the deficiencies of C Zhang's IBV scheme [20]. They point out that the assumption of ideal TPDs in each vehicle is too strong and propose to use realistic TPDs in which the vehicle's secret parameters are stored, but the system's secret parameters are not. They further point out the exorbitant costs of map-to-point-hash functions used in Zhang's scheme to map pseudonym information to a point on an elliptic curve; the CPAS scheme does not use map-to-point functions at all. They claim to have reduced the batch verification time by 18% compared to Zhang's IBV scheme. The scheme is generally similar to IBV, but pseudonyms are no longer fully generated within the TPD (since the master key no longer exists there); instead, part of the pseudonym is prestored in the TPD, then securely transmitted to a tracing manager (TRA) which subsequently generates the second part of the pseudonym and resends to the vehicle to be securely stored in the TPD. Accordingly, none but the TRA can unmask the vehicle's real identity.

In 2013, Lee et al. [24] performed a cryptanalysis of C. Zhang's IBV scheme and proved that it is vulnerable to replay attacks and does not guarantee signature non-repudiation. Replay attacks are mitigated by the simple measure of appending a timestamp to each message (which the authors of IBV implemented in their extended scheme in 2011). Non-repudiation is jeopardized, according to Lee et al., as follows: suppose a malicious vehicle generates three messages with three signatures and then swaps the signatures so that message 1 receives signature 3, for example, if the messages are verified sequentially, the signatures will be invalid, but if they are verified using Zhang's batch verification scheme, the batch will be deemed valid. In that sense, a sender can later deny having sent these messages and achieve repudiation by proving that the signatures on the individual messages do not hold out. Accordingly, Lee et al. propose a vector parameter with small-sized elements (to minimize overhead) so that the method of swapping signatures would yield an invalid outcome during batch verification. However, in order to achieve this, Lee made a minor change to the secret key generation scheme where $SK_2^i = s_2 h_2\left(ID_1^i||ID_2^i||T_i\right)$ became $SK_2^i = s_2 h_2\left(ID_1^i||ID_2^i||T_i\right)P \rightarrow SK_2^i = h_2\left(ID_1^i||ID_2^i||T_i\right)$. In that sense, Lee unknowingly rendered the signing key extractable from the signature and accordingly rendered the entire scheme vulnerable to impersonation attacks. This was pointed out by Bayat in [25] which is a response to Lee's scheme, but does not mention C. Zhang's original scheme. Incidentally, Bayat's proposed modification scheme is nothing but Zhang's original scheme with an added timestamp. Bayat also proposed to remove the vector parameter suggested by Lee without any justification, but we observe that undoing Lee's modification renders the vector parameter mathematically unusable during batch verification. The issues with Lee's scheme were also pointed out by Tzeng et al. in [26], where they also proposed an identity-based batch verification scheme that requires two bilinear pairings instead of three.

In 2020, Ali et al. [27] proposed an identity-based CPPA scheme for V2I using bilinear maps. They employ general one-way hash functions instead of map-to-point hash functions for efficiency. In their scheme, a tracing manager TRM generates pseudo-identities for vehicles and a database to trace real identities of vehicles, while the KGC generates keys for vehicles. Their batch verification scheme requires only one bilinear pairing. However,

it was pointed out in [28] that vehicles could generate their own pseudo-identities using fictitious secret values to escape accountability.

*3.2. CLAS Schemes*

Certificateless schemes are conceptually similar to identity-based schemes except for the fact that the KGC no longer has access to the full private key of each member.

### 3.2.1. History of CLAS Schemes

Proposed in [29] as a solution to the key escrow problem of identity-based cryptography (IBC), certificateless cryptography diverges from IBC in limiting the TTP's knowledge of the user's private keys to partial knowledge. In other words, the TTP creates a private key for each user based on their public key (identity string), but this key becomes a partial private key to which each user appends their own randomly generated secret value. This approach is advantageous in the sense that it still guarantees the authenticity of the participating members since the private key of each user partially comprises a TTP-issued string, but in case of an attack on the TTP, the attacker does not have access to the full private key of any user and therefore cannot decrypt any of their messages or sign messages on their behalf. The only disadvantage is that the user's public key is no longer solely a function of their identity string and the TTP's public key, so the user must publish their public key on the side since it is now affected by their privately generated key in addition to their private key. However, the public key need not be transmitted securely [29] since a part of the public key depends on the TTP's private key, hence it is verifiable at the receiver's end. There are two types of adversaries in CLAS schemes: $A_1$ is a malicious adversary which reveals the vehicle's secret key $vsk_{ID_i}$ or the vehicle's public key $vpk_{ID_i}$ or changes $vpk_{ID_i}$ but cannot reveal the master secret key of the system $msk_{KGC}$ nor obtain access to the vehicle's partial private key $psk_{ID_i}$. $A_2$ is a malicious but passive KGC who dominates the construction of the master key pair $msk_{KGC}, mpk_{KGC}$ and any vehicle's partial secret key $psk_{ID_i}$. Attacks launched by $A_1$ are called Type I attacks, and attacks launched by $A_2$ are called Type II attacks.

### 3.2.2. CLAS in the Context of VANETs

In [30,31], authors proposed protocols that utilized certificateless aggregate signatures for secure identity authentication, but they do not provide significant efficiency gains over traditional schemes due to their employment of bilinear pairings and hash-to-point operations; and they also suffer from the same high storage requirements of pseudonyms [32]. In [33], a certificateless aggregate signature scheme was proposed for V2I communication, with security against adaptively chosen message attacks in the random oracle model; however, it was later shown in [34] that the scheme is vulnerable to malicious passive KGC attacks. Furthermore, the authors in [35] showed that the same scheme cannot withstand a public key replacement attack, and the authors in [36] showed that it cannot withstand the man in the middle attack. In [37], authors achieved superior efficiency to the scheme proposed in [33] by avoiding the use of bilinear pairings, but it was shown that their scheme cannot withstand a malicious KGC attack by [35,38,39]. In [38], a full aggregation pairing-free certificateless scheme was proposed, but in [35], it was pointed out that this scheme cannot resist Type I and Type II adversary attacks. A certificateless scheme that strived towards maximizing efficiency without affecting reliability was proposed in [40]. However, it was shown in [41] that the scheme cannot withstand a malicious KGC attack, and it was also shown in [42] that the scheme cannot preserve identity privacy. The authors in [30] claimed that their certificateless scheme is more efficient than the one proposed in [40], but it was pointed out in [31,38,39] that the scheme cannot withstand a malicious KGC attack. Full aggregation certificateless schemes were also proposed in [31,38,43].

Having concluded our discussion of IBV and CLAS schemes, we provide a brief overview of the mathematical preliminaries and complexity assumptions next.

## 4. Preliminaries

In this section, we outline the theoretical underpinnings of the cryptanalyzed schemes, as well as the generic network model for VANET security schemes.

### 4.1. Elliptic Curves

We assume that $F_p$ denotes a finite field with prime order $p$, for elliptic curve $E$ with equation $y^2 = x^3 + ax + b \ mod \ p$, where $4a^3 + 27b^2 \neq 0$ and $a, b \in F_p$. We assume that $O$ denotes the point at infinity. The points of ECC make an additive group $G$ with order $q$ and generator $P$.

Point addition: Let $P$ and $S$ be two random points on ECC such that $(P, S) \in G$ where the point $P$ generates the group $G$ with large prime order $q$. When $P \neq S$, then $R = P + S$ can be computed, where $R$ denotes the intersection point of curve $E$ and line $PS$, and when $P = S$, then $R = P + S$ denotes the intersection of curve $E$ with the tangent to $E$ at $P$.

Scalar Point Multiplication: The scalar multiplication of $E$ is defined as $mP = P + P + \ldots + P$ (m times), where $m \in Z_q^*$, $m > 0$.

Elliptic Curve Discrete Logarithm Problem (ECDLP): Given two random points $P, Q \in G$ on curve E, where $Q = xP$, $x \in Z_q^*$, it has been proven difficult to calculate $x$, given $P, Q$.

### 4.2. Chinese Remainder Theorem

The Chinese Remainder Theorem (CRT) states that knowledge of the remainders of the Euclidean division of an integer $n$ by several integers facilitates the determination of the remainder of the division of $n$ by the product of these integers under the condition that they are pairwise relatively prime.

Let $k_1, k_2, \ldots, k_n$ be pairwise relatively prime positive integers such that $\gcd\left(k_i, k_j\right)_{i \neq j} = 1$ and $t_1, t_2, \ldots t_n$ be any random integers. The Chinese Remainder Theorem (CRT) states that the congruence $x \equiv t_1 \ mod \ k_1$, $x \equiv t_2 \ mod \ k_2$, $x \equiv t_3 \ mod \ k_3, \ldots, x \equiv t_n \ mod \ k_n$ has a unique solution $K$, where $K = \prod_{i=1}^{n} k_i = k_1.k_2 \ldots .k_n$.

To obtain this solution, the key server calculates $x = \sum_{i=1}^{n} t_i.K_i.K_i^{-1} mod K$, where $K_i = \frac{K}{k_i}$ and $K_i.K_i^{-1} \equiv 1 mod k_i$.

### 4.3. Network Model

The most common network architecture used to model VANET security is the two-layer model which comprises a trusted authority (TA) in the top layer and vehicles and roadside units (RSUs) in the bottom layer.

- Trusted Authority (TA): This is also known as the Trusted Third Party (TTP) or the Central Authority (CA). It is generally known to comprise a Key Generation Center (KGC) and a Tracing Authority/Tracing Manager (TRA/TRM). IBV schemes tend to combine the role of the KGC and the TRM, while CLAS schemes necessarily posit them as distinct entities. The KGC is meant to generate public and private keys (or partial keys) for all members to enable digital signature verification, while the TRM is meant to generate verifiable pseudo-identities for each member to enable traceability in case of disputes. Schemes occasionally propose redundant TAs with access to the same data repository to avert single points of failure.
- Roadside Units (RSUs): They are connected to the TA with secure wired links and to vehicles with insecure wireless connections. Different schemes assume different levels of RSU trustworthiness, where the predominant assumption is that RSUs are honest but curious. The role of RSUs varies in different schemes, from being merely a gateway to relay messages from the TA to vehicles (and back) to being a group manager that issues signing and verification keys to members within its domain and managing localized groups.
- Vehicles: Vehicles are assumed to be untrustworthy. They are equipped with onboard units (OBUs) which contain a tamper proof device (TPD). It is assumed that all vehicles

will be equipped with TPDs, but schemes diverge on the assumption of an ideal TPD (secure enough to store the master secret key of the system for self-authentication) or a realistic TPD (where only the secret key of the corresponding user is stored, and authentication is carried out elsewhere).

## 5. Zhang et al.'s Scheme

In this section, we describe Zhang et al.'s scheme [5] then perform cryptanalysis. Posited as a development to traditional IBV schemes, Zhang's scheme seeks to overcome the traditional problem of storing the master key of the system in the TPD of each vehicle by introducing a Chinese Remainder Theorem-based group key exchange. The idea is that each legitimate vehicle uses its pre-stored secret key to compute a shared group key from information regularly updated by the TA and broadcasted to vehicles via the RSUs. Having obtained the group key by computing the broadcasted value of their prestored key, vehicles then proceed to generate digital signatures on their message using both the obtained group key and their own generated random nonces. In what follows, we describe the detailed proceedings of the scheme and then suggest possible attacks.

### 5.1. Description

We detail the procedure of Zhang's scheme in Table 3.

**Table 3.** Zhang et al.'s scheme.

| Phase | Scheme | Comments |
|---|---|---|
| Setup | TA chooses large prime $p$, define $Z_p^*$ | $Z_p^*$ : cyclic group of order $p$ |
| | TA chooses large prime $q$ such that $q < \lceil \frac{p}{4} \rceil$, define $Z_q^*$ | $Z_q^*$ : cyclic group of order $q$ |
| | TA generates elliptic curve $E$ defined over finite field $F_p$ by the equation $y^2 = x^3 + ax + b \pmod{p}$ | $a, b \in F_p$ and $(4a^3 + 27b^2) \bmod p \neq 0$. We take point $P$ to be the generator of $E$ |
| | TA randomly chooses $s \in Z_q^*$ | $s$ : TA's secret key (for identity tracing) |
| | TA computes $P_{pub} = s.P$ | $P$ : generator of elliptic curve $E$ |
| | TA chooses 4 hash functions: $H_{1 \to 4} : \{0,1\}^* \to Z_q$ | |
| | TA publishes params: $(p, q, G, E, P, P_{pub}, Z_q^*, H_{1 \to 4})$ | $G = Z_p^*$ |
| GroupKey computation | TA chooses $sk_i$ from $Z_p^*$ for $n$ vehicles | $sk_i$ is the secret key of vehicle $V_i$ |
| | TA calculates $\partial_g = \prod_{i=1}^n sk_i$ | |
| | TA calculates $x_i = \frac{\partial_g}{sk_i} i = 1, 2, ..., n$ | |
| | TA calculates $y_i$ such that $x_i * y_i = 1 \bmod sk_i$ | $y_i \equiv x_i^{-1} \bmod sk_i$ |
| | TA calculates $var_i = x_i * y_i$ | |
| | TA calculates $\mu = \sum_{i=1}^n var_i$ | |
| | TA chooses small random variable $k_d \in Z_q^*$ | $k_d$ : the group key (or domain key) |
| | TA calculates $\gamma_d = \mu.k_d$ | |
| | TA signs $\gamma_d$ and lifetime $ET_i$ using $sk_{TA}$ | $sk_{TA}$ : TA's secret key (for signing) |
| | TA computes $K_{pub} = k_d.P$ | |
| | TA broadcasts: $\left\{ \gamma_d, K_{pub}, SIG_{sk_{TA}}(\gamma_d \| ET_i) \right\}$ | |
| | Members compute group key as $\gamma_d \bmod sk_i = k_d$ | Note: $k_d < q < sk_i < p$ |

**Table 3.** *Cont.*

| Phase | Scheme | Comments |
|---|---|---|
| PID generation | Driver activates TPD using fingerprint | |
| | TPD generates random nonce $r_i \in Z_q^*$ | |
| | TPD generates $PID_i = (ID_{i,1}, ID_{i,2})$ as $ID_{i,1} = r_i.P, ID_{i,2} = RID_i \oplus H_1(r_i.P_{pub})$ | Note: TA can compute: $RID = ID_{i,2} \oplus H_1(sID_{i,1})$ |
| Signing | TPD obtains $\gamma_d$ from broadcast | As mentioned in GK computation |
| | TPD computes $\gamma_d \bmod sk_i = k_d$ to obtain GK | |
| | TPD computes $\alpha_i = H_2(PID_i\|T_i)$ | Re-computable by verifier |
| | TPD computes $S_i = (\alpha_i.k_d) \bmod q$ | Re-computable by verifier (weakness) |
| | OBU inputs $M_i$ to be signed | $M_i$ : message to be broadcasted |
| | TPD computes $\beta_i = H_3(PID_i\|M_i\|T_i)$ | Re-computable by verifier |
| | TPD computes $\sigma_i = S_i + \beta_i.r_i$ | $r_i$ is only known to the signer |
| | TPD broadcasts $(M_i, PID_i, T_i, \sigma_i)$ | $T_i$ : timestamp $\sigma_i$ : signature |
| Verifying | Check freshness of $T_i$ such that $\Delta T \geq T_r - T_i$ | $T_r$ : received time; $\Delta T$ : predefined threshold (mitigates replay attacks) |
| | Check $\sigma_i.P \overset{?}{=} \alpha_i.K_{pub} + \beta_i.ID_{i,1}$ (proof: LHS: $\sigma_i.P = (S_i + \beta_i.r_i).P = S_iP + \beta_i.r_iP = \alpha_i.k_d.P + \beta_i.r_iP = \alpha_i.K_{pub} + \beta_i.ID_{i,1} \to$ RHS) | |
| Batch Verif. | Verifier received: $PID_1 \to PID_n$, $M_1 \to M_n$, $\sigma_1 \to \sigma_n$, $T_1 \to T_n$ | |
| | Check $\Delta T \geq T - T_i$ for freshness | For each message |
| | Generates random vector $v = v_1, \dots, v_n$, where $v \in [1, 2^t]$ such that $t$ is a small integer | Small exponent test |
| | Check $(\sum_{i=1}^n v_i.\sigma_i)P \overset{?}{=} (\sum_{i=1}^n v_i.\alpha_i).K_{pub} + (\sum_{i=1}^n v_i.\beta_i.ID_{1,i})$ (proof: LHS: $(\sum_{i=1}^n v_i.\sigma_i)P = \sum_{i=1}^n v_i.\sigma_i.P = \sum_{i=1}^n (v_i.\alpha_i.K_{pub} + v_i.\beta_i.ID_{i,1}) = (\sum_{i=1}^n v_i.\alpha_i).K_{pub} + (\sum_{i=1}^n v_i.\beta_i.ID_{1,i}) = RHS)$ | |

### 5.2. Cryptanalysis

In what follows, Bob is assumed to be an honest user and Oscar to be a malicious user, both are legitimate members of the same VANET. Figure 1 shows the setup and group key computation phases of the protocol, in the presence of malicious attacker Oscar.

#### 5.2.1. Impersonation Attack

Bob broadcasts $(M_B, ID_B, T_B, \sigma_B)$ to the network, and this information is received by Oscar, who is a legitimate member of the network and therefore has access to the group key $k_d$. Accordingly, Oscar computes $\alpha_B = H_2(ID_B\|T_B)$ then computes $S_B = (\alpha_B.k_d) \bmod q$ and $\beta_B = H_3(ID_B\|M_B\|T_B)$. Then, Oscar computes $r_B = \frac{\sigma_B - S_B}{\beta_B}$ and consequently computes $RID_B = ID_{B_2} \oplus H_1(r_B.P_{Pub})$.

Oscar now has access to Bob's real identity $RID_B$ and the group key $k_d$ and can impersonate Bob in future messages. Figure 2 shows the impersonation attack.

#### 5.2.2. Repudiation/Spoofing Attack

Oscar obtains $k_d$ regularly from the broadcast, generates a fake identity $RID_O^*$ and computes a corresponding pseudo-identity $PID_O^* = ID_{O,1}^*, ID_{O,2}^*$

Oscar uses $k_d$ and $PID_O^*$ in all further communications, and messages sent will be verifiable since the signature does not change if the value of $RID$ changes since each member generates the values of $PID$ based on their own generated random nonce. Even though it could be argued that the TPD prevents any member of the VANET from changing

the value of their $RID$, Oscar could still collude with Eve who is not constrained by a TPD, by giving her the group key $k_d$ and Eve can broadcast valid messages with verifiable signatures using a fake identity $PID_E^*$. Since the value of $RID$ is fake or non-existent, Oscar (or Eve) could escape retribution for any false messages sent. Figure 3 shows the repudiation attack.

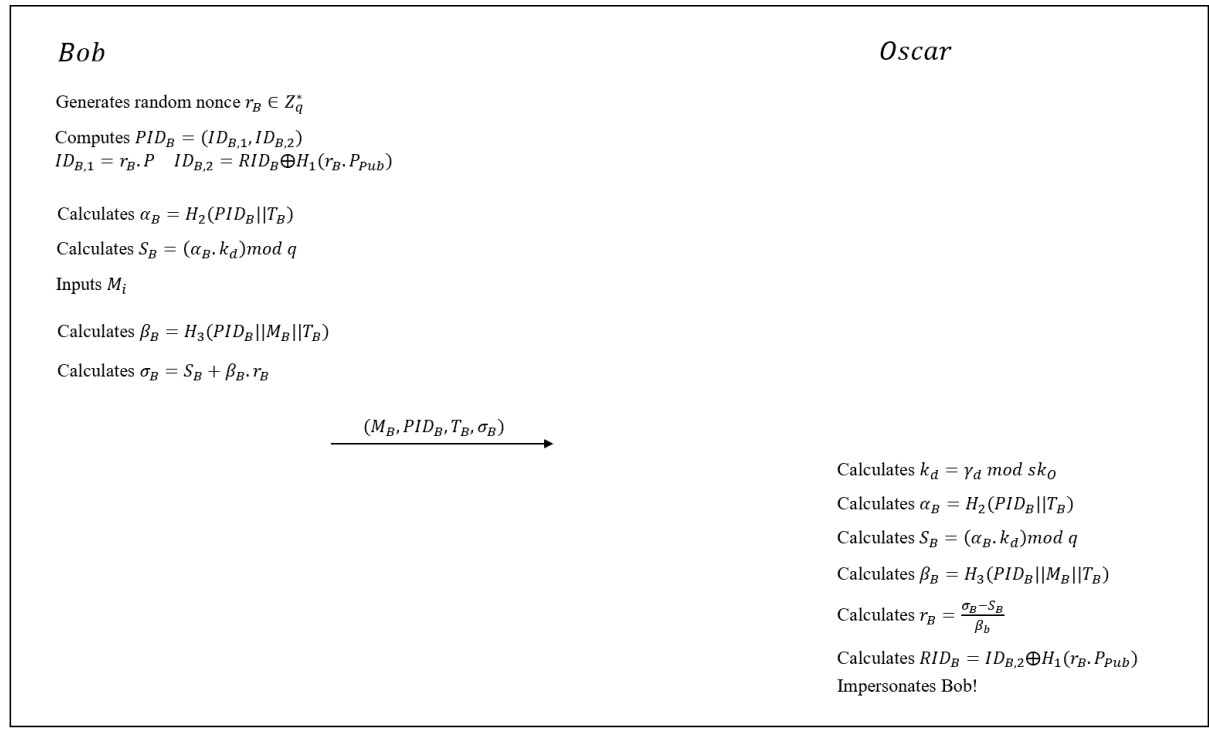

**Figure 1.** Setup and group key computation phases in Zhang's scheme.

**Figure 2.** Impersonation attack on Zhang's scheme.

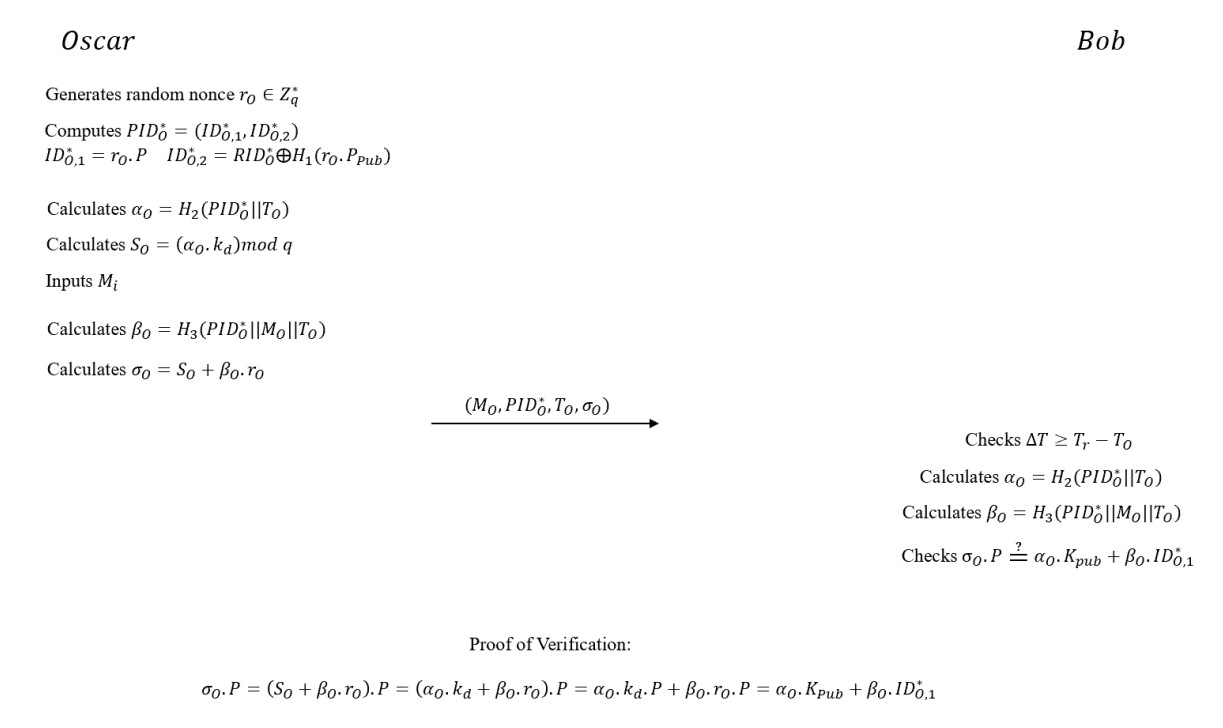

**Figure 3.** Repudiation attack on Zhang's scheme.

*5.3. Causes and Countermeasures*

In what follows, we point out the underlying causes and possible counter measures for each attack.

### 5.3.1. Impersonation Attack

The problem with Zhang's scheme is that the signature is only dependent on two private values, the group key and a random nonce, hence knowledge of the group key can immediately lead to the computation of the random nonce. Since $\sigma_i = S_i + \beta_i.r_i$, which is equivalent to $\sigma_i = (\alpha_i.k_d)\mod q + \beta_i.r_i$, and since $\alpha_i$, $\beta_i$ are hash outputs of public values (and therefore computable by all members), and $k_d$ is known to all legitimate members of the VANET, any legitimate member can obtain $r_i$ from $\sigma_i$ received from any user. In other words, given $\sigma_i$, Oscar can compute $\alpha_i$ and $\beta_i$, then they can obtain $r_i = \frac{(\sigma_i - (\alpha_i.k_d))}{\beta_i}$. If the scheme is modified so that during signing, the TPD chooses a random value $x_i \in q$ and computes $X = x_i.P$ and modifies $S_i = (\alpha_i.k_d)\mod q$ to $S_i = (\alpha_i.k_d + x_i)\mod q$, then publishes value $X$ with the broadcast, the verification equation at the receiver's end becomes $\sigma_i.P \stackrel{?}{=} \alpha_i.K_{pub} + \beta_i.ID_1 + X$.

This checks out because

$$\sigma_i.P = (S_i + \beta_i.r_i).P = (\alpha_i.k_d + x_i + \beta_i.r_i).P = \alpha_i.k_d.P + x_i.P + \beta_i.r_i.P = \alpha_i.K_{pub} + \beta_i.ID_1 + X$$

This simple modification incurs a cost of one additional elliptic point addition at the verifier's end, and Oscar can no longer compute $r_i = \frac{(\sigma_i - (\alpha_i.k_d + x_i))}{\beta_i}$ since they do not possess knowledge of $x_i$.

### 5.3.2. Repudiation Attack

The technique of generating pseudo-identities using a random nonce so that the real identity becomes irrecoverable except by the TA/TRM is optimal for solving the pseudonymity problem in VANETs. However, the scheme does not provide a method of verifying that the received pseudo-identities are indeed valid pseudo-identities. We

suggest introducing pseudo-identity verification prior to signature verification; in other words, the KGC must authenticate the identity of the user before issuing their private key. Furthermore, the TA must play a role in generating the pseudo-identities for verification to be of any consequence. Delegating the responsibility of the TA to the vehicle's TPD allows an insider member (Bob) to share the group key with an outsider (Oscar), and then, Oscar can share false information with valid self-generated credentials based on a fake identity. This highlights the importance of not relying on the TPD alone to generate a member's identity, but rather having the TA sign the identity and not allowing the member to receive a signing key from the KGC until the signature of the TA/TRM on their PID is verified.

### 5.4. A Commentary on the Cryptanalysis of This Scheme by Tao et al. [6]

In [6], it was stated that an outsider can collect different values of $\gamma_d$, then, using the equation $\gamma_d = \mu.k_d$ and the fact that $\mu$ is constant and that only $k_d$ changes, the outsider can find $\mu$ as $\mu = \gcd(\gamma_1, \gamma_2, \ldots, \gamma_n)$, the outsider can compute $k_d = \gamma.\mu^{-1}$.

We contest these claims on the grounds that $\mu$ is not in fact a constant, since $\mu = \sum_{i=1}^{n} var_i$ and every time a member joins (/leaves) the network, their corresponding $var_i$ is added (/subtracted) accordingly. Hence, $\mu$ changes with every join or leave operation. Despite the fact that it could be suggested that the scheme is still insecure because $\mu$ could be computed from successive broadcasts between which no join or leave operations occurred, we propose a simple countermeasure to that proposition. The TA can include a certain number of secret keys not given to any user in its computation of the group key, then, the TA could impose its own join/leave operations between successive broadcasts, utilizing any random subset of the fictitious secret keys to do so. In that sense, the TA could ensure that the value of $\mu$ changes between any two successive broadcasts without affecting the system performance in any significant way. Furthermore, a collusion attack between an insider and an outsider can easily achieve the same effect as Tao's proposed attack, since the insider has access to $k_d$ and can share it with an outsider without being implicated. Finally, the group key, $k_d$ is meaningless without the possession of a legitimate value of $sk_i$, and the attacker cannot create a valid signature without the latter, even if they are in possession of $k_d$.

Having described and cryptanalyzed Zhang's scheme, we proceed to describe and cryptanalyze Xiong's scheme in the following section.

## 6. Xiong et al.'s Scheme

In this section, we describe Xiong et al.'s scheme [9] and then perform cryptanalysis. Xiong's scheme is an extension of their previous scheme with modifications to resist collusion attacks and also achieve efficiency gains. We now proceed to the detailed description of the scheme followed by a cryptanalysis.

### 6.1. Description

We detail the procedure of Xiong's scheme in Table 4.

**Table 4.** Xiong et al.'s scheme.

| Phase | Scheme | Comments |
|---|---|---|
| Setup: TA = KGC + TRM | TA chooses large primes $p, q$, defines $Z_p^*$, $Z_q^*$ | $Z_p^*$, $Z_q^*$ : cyclic groups of order $p, q$ respectively |
| | TA generates elliptic curve $E$ defined over finite field $F_p$ by the equation $y^2 = x^3 + ax + b \pmod{p}$, | $a, b \in F_p$ and $(4a^3 + 27b^2) \bmod p \neq 0$. We take point $P$ to be the generator of $E$ |
| | KGC randomly chooses $s \in Z_q^*$ | $s$ : KGC's secret key |
| | KGC computes $P_{pub} = s.P$ | $P$ : generator of elliptic curve $E$ |
| | TRM randomly chooses $\alpha \in Z_q^*$ | $\alpha$ : TRM's secret key |
| | TRM computes $T_{pub} = \alpha.P$ | |

**Table 4.** *Cont.*

| Phase | Scheme | Comments |
|---|---|---|
| Setup:<br>TA =<br>KGC + TRM | TA chooses 4 hash functions : $H_i : \{0,1\}^* \to Z_q$ | |
| | TA publishes params : $\left(q, G, E, P, P_{Pub}, T_{pub}, Z_q^*, H_{1\to4}\right)$ | |
| PID Generation | $V_i$ generates random nonce $k_i \in Z_q^*$ | $VT_i$ : pseudonym validity time<br>Note: TRM can compute<br>$RID = ID_{i,2} \bigoplus H_0\left(\alpha.ID_{i,1}\|VT_i\right)$ |
| | $V_i$ generates $PID_i = (ID_{i,1}, ID_{i,2}, VT_i)$ as:<br>$ID_{i,1} = k_i.P$, $ID_{i,2} = RID_i \oplus H_0\left(k_i.T_{pub}\big\|VT_i\right)$ | |
| PPK Gen. | KGC randomly chooses $\alpha_i \in Z_q^*$ | Note : KGC does not know $RID_i$ |
| | KGC calculates $A_i = \alpha_i P$ | |
| | KGC calculates $Q_i = H_1(A_i\|PID_i)$ | |
| | $ppk_i = (\alpha_i + sQ_i) \bmod q$ | |
| | KGC transmits $(PID_i, ppk_i, A_i)$ to $V_i$ securely | $ppk_i$ : partial private key of vehicle |
| | $V_i$ checks: $ppk_i.P \overset{?}{=} A_i + Q_i.P_{Pub}$ | |
| USK Gen | $V_i$ randomly chooses $x_i \in Z_q^*$ | $x_i$ : user secret key of vehicle |
| | $usk_i = x_i\ upk_i = x_i.P_{pub}$ | |
| | $h_{i_2} = H_2(P_{ID_i}\|upk_i)$ | Re-computable by verifier |
| | $D_i = ppk_i.P_{pub}\ F_i = D_i + h_{i_2}.upk_i$ | $F_i, upk_i$ : the public key of $V_i$ |
| Signing | $V_i$ randomly chooses $u_i \in Z_q^*$ | |
| | $V_i$ computes $U_i = u_i.P$ | |
| | $h_i = H_3(M_i\|P_{ID_i}\|U_i\|t_i)$ | $M_i$ : message |
| | $T_i = h_i\left(ppk_i + h_{i_2}x_i\right) + u_i \bmod q$ | $T_i = f(u_i, M_i, P_{ID_i}, ppk_i, t_i, upk_i)$ |
| | $\delta_i = (U_i, T_i)$ | $\delta_i$ : signature |
| | $V_i$ sends: $(M_i, P_{ID_i}, F_i, upk_i, t_i, \delta_i)$ | |
| Verifying | Check freshness of $T_i$ such that $\Delta t \geq t - t_i$ | Mitigates replay attacks |
| | Check $T_i.P_{pub} \overset{?}{=} h_i F_i + U_i$ (proof : LHS : $T_i.P_{Pub} = (h_i(ppk_i + h_{i_2}x_i) + u_i).P_{Pub} = h_i ppk_i P_{Pub} + h_i h_{i_2} x_i P_{Pub} + u_i P_{Pub} = h_i D_i + h_i h_{i_2} upk_i + U_i = h_i(D_i + h_{i_2} upk_i) + U_i = h_i F_i + U_i \to$ RHS) | $h_i = H_3(M_i\|P_{ID_i}\|U_i\|t_i)$ |
| Agg. sign | Verifier received : $PID_1 \to PID_n$, $M_1 \to M_n$, $\delta_1 \to \delta_n$, $t_1 \to t_n$, $F_1 \to F_n$, $upk_1 \to upk_n$ | |
| | Check $\Delta t \geq t - t_i$ for freshness | |
| | $V_i$ randomly chooses $v \in Z_q^*$ | $V_i$ here is the receiver/verifier |
| | $V_i$ computes $pk = v.P$ | Should be $pk = v.P_{pub}$ (see Section 6.2.2) |
| | $V_i$ computes $\beta = H_4(T_1 pk\|T_2 pk\|T_3 pk \dots)$ | |
| | $V_i$ computes $T = \sum_{i=1}^{n} T_i$ | Agg. Sig. : $(T, U_1, \dots, U_n, \beta)$ |
| Batch Verify | $V_i$ computes $h_i = H_3(M_i\|P_{ID_i}\|U_i\|t_i)$ | For each $M_i, \delta_i$ pair |
| | $V_i$ computes $\varphi_i = h_i F_i + U_i$ | |
| | $V_i$ checks if $T.P_{pub} \overset{?}{=} \sum_{i=1}^{n} h_i F_i + U_i$ (proof: LHS: $T.P_{pub} = (\sum_{i=1}^{n} T_i.P_{Pub}) = \sum_{i=1}^{n} (h_i F_i + U_i) \to$ RHS)<br>$V_i$ computes $\beta = H_4(v\varphi_1\|v\varphi_2\| \dots \|v\varphi_n)$ (proof: LHS $\beta = H_4(T_1 pk\|T_2 pk\| \dots \|T_n pk) = H_4(T_1.v.P_{Pub}\| \dots \|T_n.v.P_{Pub}) = H_4(v\varphi_1\|v\varphi_2\| \dots \|v\varphi_n) \to$ RHS) | |

*6.2. Cryptanalysis*

In what follows, Oscar is assumed to be a legitimate member of the VANET with malicious intentions, Bob is also a legitimate member but with honest intentions. Figure 4 depicts the setup, pseudo-identity generation and partial-private-key generation phases of the protocol.

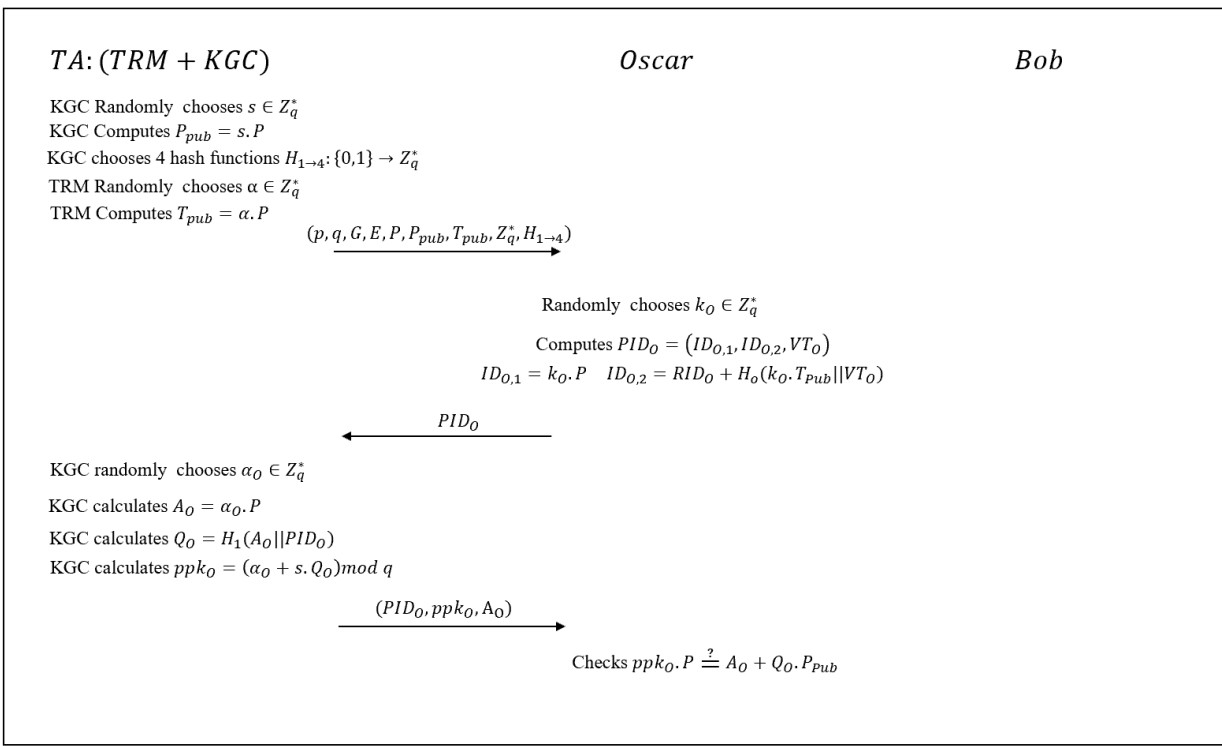

**Figure 4.** Setup, PID and PPK generation phases in Xiong's scheme.

6.2.1. Bogus Information and Repudiation Attack

Oscar honestly executes the setup phase, pseudo-identity generation phase and partial-private-key phase of the protocol. Then, Oscar generates a fake identity $RID_O^*$ and computes a new $PID_O^*$ on the basis of $RID_O^*$. Oscar also substitutes $ppk_O$ with $ppk_O^*$, which is any randomly self-generated value such that $ppk_O^* \in Z_q^*$. Oscar continues the rest of the protocol normally and inputs any fake information in the message to suit their own purpose (bogus information attack). Verification of the message is successful because the verifier uses the transmitted $PID_O^*$ to verify. Furthermore, if the message is later reported to the tracing manager, it cannot retrieve $RID_O$ since the fake identity $RID_O^*$ was used in the generation of the transmitted $PID_O^*$. The attack is depicted in Figure 5.

6.2.2. Other Flaws

Another issue with the scheme, in the signature aggregation phase, is that the verifier randomly chooses $v \in Z_q^*$ then accordingly calculates $pk = v.P$; however, the batch verification would not check out unless $pk = v.P_{pub}$. We assume that this is an unintentional mistake in the scheme, since all other random nonces in other phases were multiplied by $P_{pub}$ to obtain their corresponding public values.

*6.3. Causes and Counter Measures*

There are two main problems in Xiong's scheme. The first problem is that the TRM does not verify the generated pseudo-identities, as we indicated in our analysis of Zhang's scheme, which gives malicious users the ability to generate *PIDs* based on fictitious values of *RID*. This issue is common to both schemes. However, in Xiong's scheme, even though the generation of the *ppk* by the KGC is mathematically related to the input *PID*,

by $Q_i = H_1(A_i||PID_i)$ and $ppk_i = (\alpha_i + sQ_i)\ mod\ q$, the verifier only uses the values transmitted by the sender to verify the message, which gives the sender the ability to generate any random $ppk_i^*$ and use it to calculate subsequent parameters which leads to correct verification.

To counter these problems, we propose that the TRM validates the user-generated $PIDs$ in such a way that the KGC is first able to verify their validity before issuing a $ppk_i$. For example, an identity authentication module could be added to the scheme such that the KGC would not issue a partial private key until authenticating the TRM's signature on $PID_i$. We also propose that $Q_i = H_1(A_i||PID_i)$, based on a validated $PID_i$, be part of the verification equation. This ensures that Oscar would not be able to generate a valid signature using a fake $PID_O^*$ since he must find a value for $PID_O^*$ that satisfies the equation $H_1(A_O^*||PID_O^*) = H_1(A_O||PID_O) \rightarrow Q_o^* = Q_O$. This can only be accomplished by brute force if the hash function used is collision-resistant, which is an implicit assumption in all CPPA schemes. Finally, we propose that $D_i = ppk_i.P$ be verified separately in order to prevent Oscar from substituting $ppk_i$ with $ppk_i^*$ since this is a computationally hard problem under ECC.

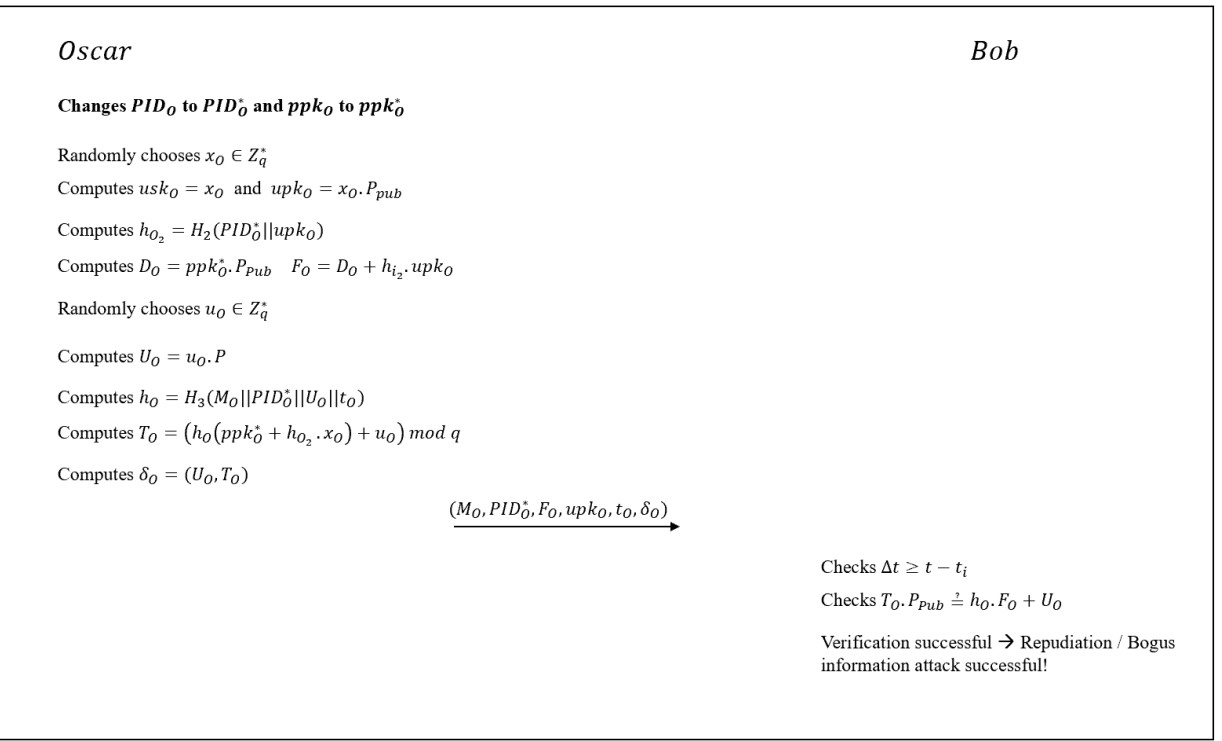

**Figure 5.** Repudiation attack on Xiong's scheme.

## 7. Conclusions and Future Work

Recently, Zhang et al. [5] and Xiong et al. [9] proposed CPPA schemes based on IBV and CLAS, respectively. We first showed that, in Zhang's scheme, an attacker Oscar can impersonate an honest user Bob by obtaining their real identity value $RID_B$ and using it to send malicious messages. We also showed that in both schemes, Oscar can generate a fictitious value $RID_O^*$ to replace their real identity $RID_O$ which would enable them to broadcast any malicious content or misleading information and escape retribution. Furthermore, in Xiong's CLAS scheme, Oscar could replace the KGC-granted partial private key with a random value of their own creation and still pass verification successfully. Furthermore, we proposed modifications in Zhang's scheme to circumvent Tao's attacks in [6]. Finally, we discussed causes and countermeasures of our cryptanalysis on both schemes. Having pointed out the vulnerabilities in both schemes, we intend to propose a CLAS scheme that employs CRT-based group key distribution without exposure to

impersonation and repudiation attacks. This will be implemented by introducing an identity authentication module to the CLAS scheme and then including the authenticated identity in the signature verification module.

**Author Contributions:** Conceptualization, writing original draft A.M.K.; review and editing, supervision H.A. and M.A. All authors have read and agreed to the published version of the manuscript.

**Funding:** This research received no external funding.

**Data Availability Statement:** Not applicable.

**Conflicts of Interest:** The authors declare no conflict of interest.

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
