# Peer review of "Cryptanalysis of Two Conditional Privacy Preserving Authentication Schemes for Vehicular Ad Hoc Networks"

_cryptography, doi:10.3390/cryptography8010004_

Round 1

Reviewer 1 Report

Comments and Suggestions for Authors

The paper systematically investigates and analyzes the security vulnerabilities of two Conditional Privacy Preserving Authentication (CPPA) schemes designed for Vehicular Adhoc Networks (VANETs). Specifically, the schemes under scrutiny are Zhang's PA-CRT and Xiong's Certificateless Aggregate Signature (CLAS). The authors embark on a comprehensive cryptanalysis endeavor, aiming to uncover potential weaknesses and enhance the robustness of these authentication mechanisms.

-The motivation for this research stems from the critical importance of securing communications in VANETs while simultaneously upholding user privacy and accountability. Zhang's PA-CRT scheme, based on identity batch verification (IBV) and the Chinese remainder theorem (CRT), is revisited following a cryptanalysis by Tao in 2023. The authors not only corroborate the findings but also propose countermeasures to fortify the scheme against identified attacks.

-In addition, the paper delves into the cryptanalysis of Xiong's CLAS scheme, which was initially proposed in 2021 and later analyzed by Shim in 2023. The authors contribute by revisiting and scrutinizing this scheme, potentially offering insights and countermeasures to address vulnerabilities identified in Shim's analysis.

-Throughout the paper, the authors provide a chronological account of the development and subsequent cryptanalysis of the targeted schemes. While exploring potential vulnerabilities, the paper aims to advance the state of knowledge in VANET security by offering solutions to fortify these Conditional Privacy Preserving Authentication schemes.

-The overall objective of the paper is to contribute to the field by not only identifying potential weaknesses but also proposing practical countermeasures to enhance the security and privacy features of the analyzed CPPA schemes in the context of VANETs.

Comments on the Quality of English Language

Need to be improved

Author Response

We thank reviewer 1 for their valued time and insights. In accordance with their commentary, we have reviewed our paper and improved it to ensure clearer presentation of the concepts and better sentence structure and grammatical fluence.  

Reviewer 2 Report

Comments and Suggestions for Authors

The idea is nice however there are few recommendations that needs to be addressed:

1.  The title requires greater clarity in alignment with the main study's design. I recommend that the authors revise the title using more appropriate and precise language.

2. More literature related to image encryption should be added i.e.,

a. Khan, J.S., Ahmad, J., Ahmed, S.S., Siddiqa, H.A., Abbasi, S.F. and Kayhan, S.K., 2019. DNA key based visual chaotic image encryption. Journal of Intelligent & Fuzzy Systems37(2), pp.2549-2561.

b. Abbasi, S.F., Ahmad, J., Khan, J.S., Khan, M.A. and Sheikh, S.A., 2019. Visual meaningful encryption scheme using intertwinning logistic map. In Intelligent Computing: Proceedings of the 2018 Computing Conference, Volume 2 (pp. 764-773). Springer International Publishing.

3. Abstract should be rewritten. The abstract should answer these questions about your manuscript: What was done? Why did you do it? What did you find? Why are these findings useful and important?

4. The method section could benefit from further improvement. It is important to provide a clear justification for the methodology approach used, explaining why it was chosen and how it is appropriate for the research question at hand. Additionally, it would be helpful to reference prior studies that have successfully used this methodology approach to strengthen the argument for its use in this particular study.

5. More statistical results should be added.

6. Limitations should be added.

7. Future work should be briefly explained.

please see the additional information below.

1. This paper basically proposed the weaknesses of the two existing studies and then proposed solution to mitigate them.

2. the study is original and relevant and addresses a specific gap.

3. More safety parameter.

4. Writing, presentation, and some additional parameters should be added as suggested in the review.

5. References could be improved.

Comments on the Quality of English Language

Few grammatical errors. It is recommended to review the manuscript after revision.

Author Response

Reply: We thank reviewer 2 for their valuable insights and comments. We shall address each comment individually.

1) “The title requires greater clarity in alignment with the main study’s design”.

The title of our paper is: “Cryptanalysis of two conditional privacy preserving authentication schemes for VANETs.” The title clearly reflects the contents of the paper, since the objective of the paper is to cryptanalyze two conditional privacy preserving authentication schemes for VANETs.

2) “More literature related to image encryption should be added”.

The scope of our paper is the cryptanalysis of two privacy preserving authentication schemes for vehicular adhoc networks. In order to preserve the scope, we have exclusively chosen literature discussing digital signature schemes in vehicular adhoc networks.   

3) “Abstract should be rewritten. It should answer these questions: What was done, why did you do it, what did you find? Why are these findings useful or important?”

This is our original abstract:

“Conditional Privacy Preserving Authentication (CPPA) schemes are an effective way of securing communications in Vehicular Adhoc Networks (VANETs), as well as ensuring user privacy and accountability. Cryptanalysis plays a crucial role in pointing out the vulnerabilities in existing schemes to enable the development of more resilient ones. In 2019, Zhang proposed a CPPA scheme for VANET security (PA-CRT), based on identity batch verification (IBV) and Chinese remainder theorem (CRT). In 2023, this scheme was crypt-analyzed by Tao. In this paper we provide our own cryptanalysis of Zhang’s scheme and propose counter measures to mitigate Tao’s attacks. Furthermore, in 2021, Xiong proposed a Certificateless Aggregate Signature (CLAS) scheme which was cryptanalyzed by Shim in 2023. Xiong subsequently proposed a modified CLAS scheme which is also cryptanalyzed in this paper.”

This is our new abstract:

“Conditional Privacy Preserving Authentication (CPPA) schemes are an effective way of securing communications in Vehicular Adhoc Networks (VANETs), as well as ensuring user privacy and accountability. Cryptanalysis plays a crucial role in pointing out the vulnerabilities in existing schemes to enable the development of more resilient ones. In 2019, Zhang proposed a CPPA scheme for VANET security (PA-CRT), based on identity batch verification (IBV) and Chinese remainder theorem (CRT). In this paper we cryptanalyze Zhang’s scheme and point out its vulnerability to impersonation and repudiation attacks. In 2023, Zhang’s scheme was cryptanalyzed by Tao, however we point out flaws in Tao’s cryptanalysis due to invalid assumptions; hence we propose countermeasures to Tao’s attacks. Furthermore, in 2021, Xiong proposed a Certificateless Aggregate Signature (CLAS) scheme which is also cryptanalyzed in this paper. Finally, we analyze the causes and countermeasures by pointing out the vulnerabilities in each scheme that enabled us to launch successful attacks, and proposing changes that would fortify these schemes against similar attacks in the future.”

We hope this new abstract effectively answers the questions posed by reviewer 2:

  1. A) What was done?

1) “In this paper we cryptanalyze Zhang’s scheme and point out its vulnerability to impersonation and repudiation attacks”

2) “We point out flaws in Tao’s cryptanalysis due to invalid assumptions; hence we propose countermeasures to Tao’s attacks”

3) “Xiong proposed a Certificateless Aggregate Signature (CLAS) scheme which is also cryptanalyzed in this paper”

  1. B) Why did you do it?

“Cryptanalysis plays a crucial role in pointing out the vulnerabilities in existing schemes to enable the development of more resilient ones.”

  1. C) What did you find?

We found vulnerabilities in both schemes which are referred to more clearly in the new abstract.

  1. D) Why are the findings useful?

To increase the resilience of these schemes and other schemes in the future.

4) “The method section could benefit from further improvement. It is important to provide clear justification for the methodology approach used…”

We enthusiastically support the importance of providing clear justification for the choice of methodology approach in any research paper. In context of our cryptanalysis, we have sought to make explicit the vulnerabilities in both schemes that enabled us to launch successful attacks. We encourage the reviewer to check our amended sections (5.3 and 6.3) which we have rewritten to more clearly communicate the logic and theory behind our cryptanalysis.

5) More statistical results should be added.

We have carefully considered the suggestion of adding more statistical results to our paper. While we understand the importance of statistical analysis in some contexts, our focus in this work lies primarily in the field of cryptanalysis, aiming to identify vulnerabilities in existing Conditional Privacy Preserving Authentication (CPPA) schemes for VANETs.

Given the nature of our study, we believe that expanding on statistical results may not align with the primary objectives and contribution of our work. Our intent is to provide a comprehensive cryptanalysis of the proposed schemes and to propose effective countermeasures.

6) Limitations should be added

We acknowledge the importance of addressing the limitations of our study to enhance the clarity and completeness of our contributions. Upon careful consideration, we would like to highlight the following limitations in our current work:

Scope of Cryptanalysis: Our focus primarily centers on the cryptanalysis of specific Conditional Privacy Preserving Authentication (CPPA) schemes proposed by Zhang and Xiong. While we aim to provide a thorough examination of these schemes, it is important to note that the findings and proposed countermeasures are specific to the analyzed schemes.

Generalization to Other Schemes: The conclusions drawn from our cryptanalysis may not be directly generalizable to all CPPA schemes in Vehicular Adhoc Networks (VANETs). Each scheme may have unique characteristics and vulnerabilities that require individual scrutiny.

Assumptions and Model Limitations: Our analysis is based on certain assumptions and models inherent to the studied schemes. Deviations from these assumptions or variations in real-world scenarios may impact the applicability of our findings.

7) Future work should be briefly explained

Original discussion of future work:

“Having pointed out the vulnerabilities in both schemes, we intend to propose a CLAS scheme that employs CRT based group key distribution without exposure to impersonation and repudiation attacks.”

Modification of future work based on the reviewer’s comments:

“Having pointed out the vulnerabilities in both schemes, we intend to propose a CLAS scheme that employs CRT based group key distribution without exposure to impersonation and repudiation attacks. This will be implemented by introducing an identity authentication module to the CLAS scheme and then including the authenticated identity in the signature verification module.” 

Reviewer 3 Report

Comments and Suggestions for Authors

The paper provides cryptanalysis for two conditional privacy-preserving authentication schemes for VANETs. The paper is interesting but some issues need to be solved.

1.       The information provided inside the abstract needs to be more fluently, clearly, and concisely presented. Generally, the abstract is not a “related work”-like section, so mentioning Tao’s and Shim’s approaches in the abstract section can be omitted. Moreover, there are two typos, namely “crypt-analyzed” instead of “cryptanalyzed” (line 15), and “crypt analysis” instead of “cryptanalysis” (line 16).

2.       Lines 26-28: the sentence “A C-ITS is composed…” is badly constructed and needs to be reshaped;

3.       Line 67: why the reference [1] was included there?

4.       Table 1: there are no definitions there, only notations; A definition means something else.

5.       Lines 235-236 need to be deleted, since the table has its own title.

6.       Table 2 and Table 3 need at least one line between column headers and the rest of the tables.

7.       In Table 4, right column, an error is marked with boldface “ERROR: should be pk=…”. Why?

8.       The countermeasures proposed for the two schemes need to be explained and theoretically tackled in their underlying assumptions and implications. Please provide more details.

Comments on the Quality of English Language

English needs minor polishing.

Author Response

We thank reviewer 3 for their invaluable comments. We shall reply to each comment individually.

1) The information provided inside the abstract needs to be more fluently, clearly, and concisely presented. Generally, the abstract is not a “related work”-like section, so mentioning Tao’s and Shim’s approaches in the abstract section can be omitted. Moreover, there are two typos, namely “crypt-analyzed” instead of “cryptanalyzed” (line 15), and “crypt analysis” instead of “cryptanalysis” (line 16).

This is our original abstract:

“Conditional Privacy Preserving Authentication (CPPA) schemes are an effective way of securing communications in Vehicular Adhoc Networks (VANETs), as well as ensuring user privacy and accountability. Cryptanalysis plays a crucial role in pointing out the vulnerabilities in existing schemes to enable the development of more resilient ones. In 2019, Zhang proposed a CPPA scheme for VANET security (PA-CRT), based on identity batch verification (IBV) and Chinese remainder theorem (CRT). In 2023, this scheme was crypt-analyzed by Tao. In this paper we provide our own crypt analysis of Zhang’s scheme and propose counter measures to mitigate Tao’s attacks. Furthermore, in 2021, Xiong proposed a Certificateless Aggregate Signature (CLAS) scheme which was cryptanalyzed by Shim in 2023. Xiong subsequently proposed a modified CLAS scheme which is also cryptanalyzed in this paper.”

 This is our new abstract:

“Conditional Privacy Preserving Authentication (CPPA) schemes are an effective way of securing communications in Vehicular Adhoc Networks (VANETs), as well as ensuring user privacy and accountability. Cryptanalysis plays a crucial role in pointing out the vulnerabilities in existing schemes to enable the development of more resilient ones. In 2019, Zhang proposed a CPPA scheme for VANET security (PA-CRT), based on identity batch verification (IBV) and Chinese remainder theorem (CRT). In this paper we cryptanalyze Zhang’s scheme and point out its vulnerability to impersonation and repudiation attacks. In 2023, Zhang’s scheme was cryptanalyzed by Tao, however we point out flaws in Tao’s cryptanalysis due to invalid assumptions; hence we propose countermeasures to Tao’s attacks. Furthermore, in 2021, Xiong proposed a Certificateless Aggregate Signature (CLAS) scheme which is also cryptanalyzed in this paper. Finally, we analyze the causes and countermeasures by pointing out the vulnerabilities in each scheme that enabled us to launch successful attacks, and proposing changes that would fortify these schemes against similar attacks in the future.”

We are grateful to Reviewer 3 for pointing out the typos, we hope this new abstract provides a clearer depiction of our contribution.  

2) Sentence in lines 26-28 is badly constructed and needs to be reshaped.

Original Sentence: A C-ITS is composed of vehicles, roadside infrastructure and a traffic management authority, communication takes place in an ad-hoc manner through self-organized networks called vehicular ad-hoc networks (VANETs).

Modified Sentence: A C-ITS is composed of vehicles, roadside infrastructure and a traffic management authority. Therein, communication takes place in an ad-hoc manner through self-organized networks called vehicular ad-hoc networks (VANETs).

3) Line 67: why was reference [1] included here

This was a typo and has been removed, we thank you for pointing it out.

4)Table 1, there are no definitions here, only notations. A definition means something else.

We agree with reviewer 3 and have accordingly changed the word “definition” into the word “term”.

5) Lines 235-236 need to be deleted since the table has its own title.

This redundancy was not intentional and has been removed. We thank reviewer 3 for their meticulousness and valuable insights.

6) Adding a line between table headers and table contents in tables 2 and 3.

Agreed and implemented.

7) In Table 4, right column, an error is marked with boldface “ERROR: should be pk=…”. Why?

This is meant to outline that the author made a mathematical error here (that would not check out during verification), part of our cryptanalysis included rectifying this error as was outlined in section 6.2.2 “other flaws”. Accordingly, we have removed the work “Error” as it might be misleading, and we have instead prompted the reader to refer to the proper section. We apologize for the confusion and thank the reviewer for pointing it out.

8) The countermeasures proposed for the two schemes need to be explained and theoretically tackled in their underlying assumptions and implications. Please provide more details.

We have modified the corresponding sections (5.3 and 6.3) to provide more details that hopefully better explain the logic behind our proposed improvements.

Reviewer 4 Report

Comments and Suggestions for Authors

This paper is good - only minor errors in proofreading.   For example, sometimes cryptanalysis is split into two words "crypt analysis" such as line 16 and line 61.

The paper "Cryptanalysis of two Conditional Privacy-Preserving Authentication schemes for VANETs" provides a comprehensive analysis of two schemes aimed at securing communications in Vehicular Adhoc Networks (VANETs) and ensuring user privacy and accountability. The first scheme, proposed by Zhang, is a Conditional Privacy-Preserving Authentication (CPPA) scheme based on identity batch verification (IBV) and the Chinese remainder theorem (CRT). The second scheme, proposed by Xiong, is a Certificateless Aggregate Signature (CLAS) scheme. The paper highlights vulnerabilities in both schemes, including impersonation attacks, repudiation attacks, and bogus information attacks, and proposes countermeasures to mitigate these vulnerabilities. Additionally, the paper discusses the implications of cryptanalysis for the future of VANET security and emphasizes the importance of developing more resilient schemes.

Comments on the Quality of English Language

Ok except for minor proofreading such as lines 16 and line 61 split cryptanalysis into two words.

Author Response

We thank reviewer 4 for their warm comments and motivating response.

Round 2

Reviewer 1 Report

Comments and Suggestions for Authors

In reviewing the manuscript, I am pleased to state that the authors have diligently and comprehensively addressed all of my comments. Their responsiveness and attention to detail are commendable, as they have taken into account each suggestion and critique provided during the review process. The revisions made by the authors have significantly improved the clarity, coherence, and overall quality of the work. Notably, they have successfully incorporated additional evidence, refined their arguments, and ensured the thoroughness of their explanations. As a result, the manuscript now stands as a more polished and robust contribution to the field, ready for publication. I commend the authors for their dedication to enhancing the manuscript and believe that their efforts will be well-received by the academic community.

Comments on the Quality of English Language

Minor improvements required

Reviewer 2 Report

Comments and Suggestions for Authors

All comments have been addressed!!!